# HumanLiker: A Human-like Object Detector to Model the Manual Labeling Process

**Haoran Wei**[1,*]  **Ping Guo**[2]  **Yangguang Zhu**[1]  **Chenglong Liu**[1]  **Peng Wang**[2]

[1]University of Chinese Academy of Sciences  [2]Intel Labs China

{weihaoran18, zhuyangguang19, liuchenglong20}@mails.ucas.ac.cn

{ping.guo, patricia.p.wang}@intel.com

## Abstract

Popular object detection models generate bounding boxes in a different way than we humans. As an example, modern detectors yield object box either upon the regression of its center and width/height (center-guided detector), or by grouping paired estimated corners (corner-guided detector). However, that is not the pattern we manually label an object due to high degrees of freedom in searching centers or low efficiency of grouping corners. Empirically, humans run two steps to locate an object bounding box manually: 1) click the mouse at the top-left corner of object, and then drag the mouse to the bottom-right corner; 2) refine the corner positions to make the bounding box more precisely, if necessary. Inspired by this manual labeling process, we propose a novel human-like detector, termed as HumanLiker, which is devised as a two-stage end-to-end detector to simulate the two aforementioned. Like we humans in manual labeling, HumanLiker can effectively avert both the thorny center searching and heuristic corner grouping. Different from the mainstream detector branches, *i.e.*, the center/corner-guided methods, the Human-Liker provides a new paradigm which integrates the advantages of both branches to balance the detection efficiency and bounding box quality. On MS-COCO test-dev set, HumanLiker can achieve 50.2%/51.6% and 53.8%/55.6% in term of AP with ResNeXt-101 and SwinTransformer backbones in single/multi-scale testing, outperforming current popular center/corner-guided baselines (*e.g.*, DETR/CornerNet) by a large margin, with much less training epochs and higher inference FPS. Code will be available at https://github.com/Ucas-HaoranWei/HumanLiker.

## 1 Introduction

Object detection is an active research direction in artificial intelligence, aiming to know "where are objects of interest" in an image. In almost a decade, with the development of deep learning [15, 13] techniques, object detectors have been developed rapidly and achieved promising performance in terms of both accuracy and efficiency, which enables emerging applications in various fields, *e.g.*, driver assistance [31], ID card identification [18], image search, *etc*.

Among modern object detectors, the most popular branch is the center-guided type [27, 24, 20, 1, 36, 30, 3] which often regards object centers as reference points to predict the object location and size (width/height). Another interesting branch is the corner-guided type [14, 7, 6, 32] which determines object boundaries via estimating and grouping corner pairs. These two detection paradigms enjoy their distinctive advantages while enduring the shortcomings. More specifically, center-guided methods are effective in estimating the coarse object locations to achieve higher $AP_{50}$ compared to corner-guided methods that suffer weak robustness corner grouping. For instance, despite the

---

*This work was done when the first author was interning at Intel Labs China.

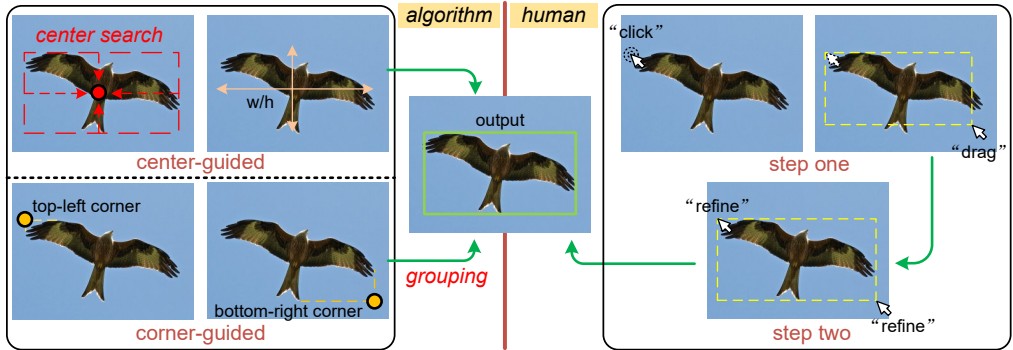

Figure 1: A comparison in generating object box of modern detectors with our humans. The center locating and corner grouping ("red" italic) are laborious and unnecessary for manual labeling.

AP (mean of $\mathrm{AP}_{(50:5:95)}$) of Faster R-CNN [27] is lower than that of CornerNet [14] (36.2% *vs.* 40.6%), the $\mathrm{AP}_{50}$ of the former is still higher than the latter (59.1% *vs.* 56.4%). In contrast, due to the fact that a corner is with lower degrees of freedom than a center (a corner needs two boundaries *vs.* a center requires four boundaries, as shown in Figure 1), corner-guided detector is potential to achieve higher-quality bounding box, especially for objects with peculiar geometry. As an example, the $\mathrm{AP}_{90}$ of CornerNet is 3.9% higher than that of Cascade R-CNN [1] (23.4% *vs.* 19.5%) under on-par accuracy (40.6% *vs.* 41.1% AP). Accordingly, it is tempting to deliberate: *Can we combine advantages of the above two detection branches?* We believe the answer is affirmative since that is actually what we humans do when we manually label object boundaries.

Intuitively, when humans desire to label an object bounding box manually, the process is usually divided into two steps to ensure high qualities. As a first step, we generally pinpoint the top-left corner of a bounding box via the top-most and left-most of an object. Typically, we are used to click the mouse at the position of top-left corner and then drag the mouse to a location of the bottom-right corner, as illustrated in Figure 1. In such step, the "click" and "drag" correspond to the corner estimation and box regression processes of corner/center-guided detectors, respectively. However, it is hard to locate precisely corners visually at once location. So we need a second step to adjust the corner coordinates to get precise bounding boxes. Compared to modern object detectors illustrated in Figure 1, the manual labeling strategy is more elegant without ambiguous center localization or complex corner grouping.

Inspired by the manual labeling process, a human-like detector, termed as HumanLiker, is proposed in this paper. HumanLiker is a two-stage detector simulating two steps in manual labeling. In the first stage, HumanLiker models the "click" and "drag" mouse operations to pinpoint corners and decode coarse object boxes based on three output maps, *i.e.*, a heatmap to estimate top-left corner, an offset map to refine the corner, and a distance map to regress the relative distance from bottom-right corner to top-left corner. Compared to popular corner-guided works [14, 7, 6], the first stage of our HumanLiker has three main differences: 1) the HumanLiker only estimates one corner, *i.e.*, the top-left corner, to avoid the time-consuming corner grouping process; 2) Instead of single-level feature, we use multi-level features based on FPN [19] to better fit object corners with different size and context; 3) Our model no longer relies on elaborate network structure friendly for corner estimations, such as the Hourglass [23], it is robust to various backbones such as ResNet [11] or its variant [10]. In the second stage, we extract RoIs (Region of Intersect) upon box proposals generated in the first stage to refine the top-left and bottom-right corners just like we humans in manual labeling. Different from traditional two-stage detectors [27, 1], the refinement references of HumanLiker are the top-left coordinates instead of the center points. This is because the strong prior knowledge that a corner is easier to be located than a center is one-size-fits-all, no matter for we humans and a detection network (see Section 4.5 for details). As mentioned above, the construction process of HumanLiker is neat and bionic, which makes the model intelligible and friendly to follow.

We select the challenging MS-COCO [21] benchmark to verify the effectiveness of the proposed HumanLiker. As a new detection paradigm, HumanLiker greatly exceeds classic baselines (*e.g.*, Faster R-CNN and CornerNet) and even achieves competitive accuracy compared to latest state-of-

the-arts. Specifically, it can yield AP of 50.2% and 53.8% in single-scale testing as well as 51.6% and 55.6% in multi-scale upon ResNeXt-101 [10] and SwinTransformer [22] backbones, respectively. Besides, HumanLiker achieves good efficiency-accuracy balance. Only needing 12 epochs to train, it can yield an AP of 47.1% at an 18 FPS on COCO val2017 set, which is much more efficient than most of advanced methods. More importantly, we believe that the HumanLiker still enjoys much improvement room, *e.g.*, further optimizing top-left corners location and long-range distances regression. Accordingly, we believe the powerful and intelligible HumanLiker can turn into a popular and excellent baseline in the further and we encourage researchers to rethink the construction way of object detector from a bionic perspective.

## 2 Related Work

### 2.1 Center-guided Detector

Center-guided detector is the most active branch in object detection community, which usually defines a sea of center locations (points/areas) as positive samples to directly regress the heights and widths of objects. We classify detectors into this category as long as they directly predict object centers.

Faster R-CNN [27] applies center-driven anchors to guide the proposal regression and achieves end-to-end trainable. Afterwards, the center-guided anchor mechanism becomes a de-facto standard in two-stage detectors. To improve the detector efficiency, a few center-guided one-stage algorithms [24–26] also be devised. They delete the RPN and directly run regression and classification at anchor centers. RetinaNet [20] introduces the Focal loss to balance the hard and easy samples to boost its accuracy. Different from the above anchor-based detectors, FCOS [30] and DETR [3] present two types of center-guided anchor-free manners. FCOS models lots of pixels within center areas of object as positives, and then predicts four vectors directly at each positive location without the guidance of an anchor. DETR utilizes the Hungarian algorithm to match predicted centers with ground-truth ones.

Although center-guided detector has achieved great success, they also face shortages, *e.g.*, pinpoint the center of a box is arduous actually — generally needing all four boundaries of the corresponding object as reference.

### 2.2 Corner-guided Detector

Instead of predicting center points directly, corner-guided detector estimates and extracts corner keypoints upon heatmaps to decode object boxes. Due to avoiding the direct search for the center point, we argue this corner-guided detector is more bionic.

CornerNet [14] outputs object boxes via estimating and grouping corner pairs. The grouping method in CornerNet is that if a top-left corner and a bottom-right corner belong to the same object, the distance between their embedding vectors will be small [14] . To address mismatched corners across different objects, CenterNet [7] introduces an extra prediction head, making the corners grouping become triplets grouping. CentripetalNet [6] abandons the 1D pull-push embeddings and presents a centripetal grouping method with a 2D-embedding form to better group paired corners. Besides, CPN [8] removes mismatched corners via sub-networks to further lift accuracy. As described above, grouping process is an obstacle to improve the corner-guided detector, which is also an unnecessary step for humans in labeling.

We devise HumanLiker as a new human-like detection branch that combines strong points of both center/corner-guided branches to further explore the potential of deep-network-based detectors.

## 3 Method

### 3.1 Towards Human-like Detection Model Construction

As stated in Section 1, there are usually two steps in the manual labeling process. Here we further dissect how to model each step based on networks along with Figure 1 and 2. Given an input image $I^{H \times W}$ with objects to be labeled, we humans usually "click" our mouse at the top-left corner location of the interested object to pinpoint its position at first. In HumanLiker, we utilize keypoint estimation method to model this procedure. As illustrated in Figure 2, we adopt a heatmap to predict and

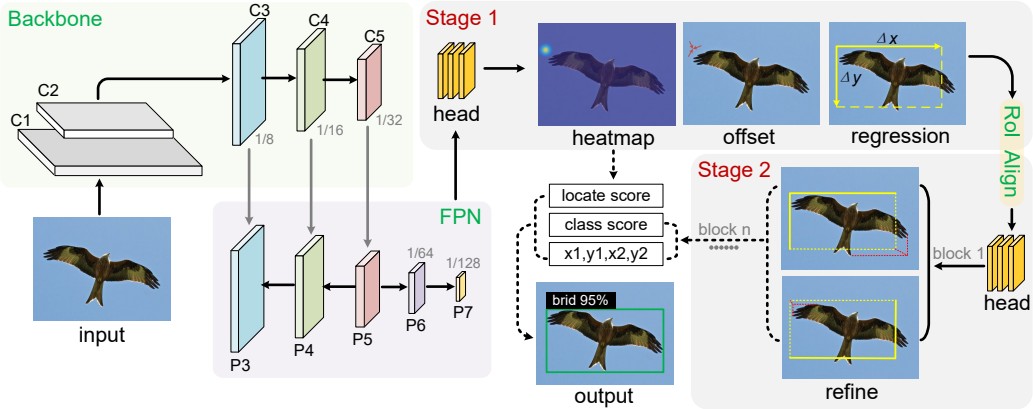

Figure 2: The structure of HumanLiker. HumanLiker is a two-stage detector corresponding to two steps of manual labeling. In the first stage, we utilize heatmap and regression map to simulate the "click" and "drag" mouse operations. In the second stage, we use cascade blocks to gradatim refine the bottom-right and top-left corners, which is very like our humans in labeling.

pinpoint the top-left corner. After locating the top-left corner, the next procedure is to "drag" the mouse from the located top-left position to the bottom-right one. We apply distance regression to output the route of "drag". To be a little specific, at each top-left location, we regress $\Delta x$ and $\Delta y$ pairs to point to the bottom-right coordinate. Empirically, it is hard for us to label the bounding box precise enough by once location and we generally need a second step to adjust the box boundary. This is also what to be done in the second stage of HumanLiker. In the stage two, we first extract RoI features upon proposal boxes generated in the first stage. Then we use $n$ cascaded blocks to gradatim refine the bottom-right corner locations as well as the top-left ones. We do not follow traditional RoI heads [27, 1] that adjust centers because optimizing corners is more direct and bionic. In what follows, we will delve into each module of the proposed HumanLiker.

## 3.2 Backbone and FPN

Figure 2 illustrates the overall structure of the proposed method. We can see that HumanLiker is a neat and unified network composed of a backbone, an FPN, and several detection heads. Like most of detection algorithms, the backbone/FPN act as extracting/building multi-level features with rich information from input images. We extract C3-C5 feature maps in the backbone and then input them into FPN. The FPN yields P3-P7 feature maps with different strides ($\times 8, \times 16, \times 32, \times 64, \times 128$) via tiny convolution modules to fit objects of all sizes. The allocation strategy of object-sizes in HumanLiker follows previous methods [30, 35]. It is worth to notice that multi-level prediction is significant for HumanLiker due to long-range regression from the top-left corner to the bottom-right.

## 3.3 The First Stage of HumanLiker

As stated in Section 3.1, the first stage of HumanLiker is devised to simulate the rough annotation process of manual labeling, *e.g.*, the "click" and "drag" mouse operations. As shown in Figure 2, there are three outputs in such stage and we will elaborate their roles and design methods in what follows.

### 3.3.1 Locate top-left corner via heatmap

We employ heatmap to detect the top-left corner keypoint. For each FPN feature with the size of $\frac{H}{S} \times \frac{W}{S}$, HumanLiker predicts a heatmap with the same size to estimate top-left corner keypoints for objects with different sizes via four tiny convolution layers. Here, $H$ and $W$ are the height and width of an input image. $S$ denotes the output stride of FPN. Each pixel value $y \in [0,1]^{\frac{H}{S} \times \frac{W}{S}}$ in heatmap represents the confidence of being judged as a top-left corner.

For each corner target, there is only one true positive sample location with the value set to 1, and all other locations are negative samples with values set to 0. To reduce the penalty [14] given to negative

locations which are close to the positive as well as balance the positive/negative samples to some extent, we adopt a Gaussian kernel (Gk) to generate a pseudo label [14] for each ground-truth corner:

$$\text{Gk} = \exp(-\frac{(x - x_m)^2 + (y - y_m)^2}{f(\text{object size})^2}) \tag{1}$$

where $(x, y)$ is a coordinate in the pseudo ground-truth map. $(x_m, y_m)$ is an object top-left corner mapped in a grid center of FPN. $f(\cdot)$ is an object-size adaptive function [2, 14, 35]. Through the above design, each pixel value in the pseudo label map can reflect the distance information representing how far from a real corner location, which is regarded as the amount of penalty reduction. In the training phase, we utilize a distance-penalty-aware and class agnostic Focal loss [20] as the objective:

$$\mathcal{L}_h = -\frac{1}{N} \sum_{xy} \begin{cases} (1 - Z'_{xy})^\alpha \log Z'_{xy}, & \text{if } Z_{xy} = 1 \\ (1 - Z_{xy})^\beta Z'_{xy}{}^\alpha \cdot \log(1 - Z'_{xy}), & \text{otherwise} \end{cases} \tag{2}$$

where $\alpha$ and $\beta$ are two hyper-parameters, $\alpha$ is utilized to adjust the weights of easy/hard samples, and $\beta$ acts as adjusting the distance penalty reduction. The values of $\alpha$ and $\beta$ are fixed to 2 and 4 respectively, following CornerNet [14]. $Z'_{xy}$ represents the predicted pixel value at a coordinate $(x, y)$ in the corresponding heatmap and $Z_{xy}$ corresponds to the pseudo-ground-truth as Eq.1. $(1 - Z_{xy})$ is the distance penalty reduction. $N$ is the number of objects.

It is worth noting that the value of each pixel in predicted heatmap can also represent a top-left corner locate score and that locate score also takes part in forming the final box score in HumanLiker.

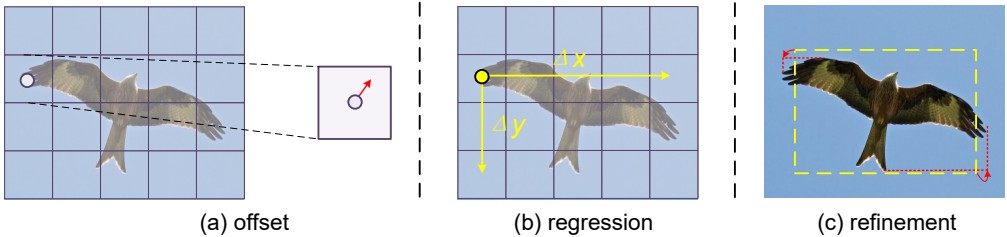

|     (a) offset     |     (b) regression     |     (c) refinement     |

Figure 3: Some details of HumanLiker. (a) To alleviate the discretization error in the process of remapping corners, we regress an offset at each top-left corner location mapped in the corresponding FPN grid center. (b) At each mapped top-left location, we regress the relative distance $(\Delta x, \Delta y)$ of bottom-right corner from the top-left. (c) HumanLiker further refine the top-left locations and adjust the relative distances between top-left and bottom-right corners in the second stage.

### 3.3.2 Adjust extracted top-left corner location via offsets

We need to extract top-left coordinates in multi-level heatmaps by *top-k* [14] method and remap the coordinates to originally input image. Due to the fact that there are output strides in FPN, it may cause discretization errors in the process of remapping corners. We address this potential problem by introducing offset [14]. To be specific, we regress offsets at each mapped top-left corner location in the corresponding FPN grid center, as shown in Figure 3 (a). With the above description, each ground-truth offset $o(\delta x, \delta y)$ is defined as:

$$o(\delta x, \delta y) = (x_m - x^*, \ y_m - y^*) \tag{3}$$

where $(x^*, y^*)$ is a real top-left corner coordinate. $(x_m, y_m)$ is a ground-truth location (a mapped grid center of FPN). During training, we adopt the smooth L1 [27] loss to supervise the offset:

$$\mathcal{L}_o = \frac{1}{N} \sum_{i=1}^{N} \text{smooth L1}(o_i, \hat{o}_i) \tag{4}$$

where $N$ is the number of positives. $o_i$ and $\hat{o}_i$ are the ground-truth and predicted offsets, respectively. Benefited by the predicted $\hat{o}(\delta x, \delta y)$, we can adjust the original top-left corner coordinate $p(x, y)$ as:

$$p(x, y)_a = p(x, y) - \hat{o}(\delta x, \delta y) \tag{5}$$

where $p(x, y)_a$ is the new top-left corner location after adjustment upon the corresponding offset.

### 3.3.3 Predict bottom-right corner location via distance regression

To model the "drag" mouse procedure in manual labeling, HumanLiker predicts the relative location for each bottom-right corner at the corresponding top-left corner location. As illustrated in Figure 3 (b), like the offset prediction, the model also adopts distance regression at the mapped grid center of the top-left corner in FPN. The regression targets are the coordinate differences $(\Delta x, \Delta y)$ between two corners, which is actually the width and height of the corresponding object.

$$(\Delta x, \Delta y) = (x_{br} - x_m, \ y_{br} - y_m) \tag{6}$$

where $(x_{br}, \ y_{br})$ is the coordinate of a bottom-right corner. The definitions of $(x_m, y_m)$ is the same as Eq. 3. In this part, we utilize the GIoU [28] loss as objective for training, aiming to supervise objects of different sizes with equal intensity.

$$\mathcal{L}_r = \frac{1}{N} \sum_{i=1}^{N} \mathrm{GIoU}(box(\Delta x, \Delta y), box(\hat{\Delta x}, \hat{\Delta y})) \tag{7}$$

where $(\hat{\Delta x}, \hat{\Delta y})$ are predicted relative distance of the bottom-right corner and $box(\cdot)$ represents a box constructed by the corresponding top-left/bottom-right corners. After acquiring the regressed $(\Delta x, \Delta y)$, we can decode the predicted bottom-right corner location $p(x_{br}, y_{br})$ directly as:

$$p(x_{br}, y_{br}) = p(x, y) + (\Delta x, \Delta y) \tag{8}$$

Along with the coordinates of paired corners, the proposal box ($P_{box}$) utilized to input the second stage is defined as:

$$P_{box} = [p(x, y)_a, p(x_{br}, y_{br}), s(x, y)] \tag{9}$$

where $p(x, y)_a$ is the adjusted top-left corner location. $s(x, y)$ is the confidence of this corner, which can be extracted in heatmap and regarded as a locate score of the top-left corner.

### 3.4 The Second Stage of HumanLiker

The purpose of the second stage of HumanLiker is to refine the top-left corner coordinate further as well as adjust the bottom-right one like the fine trimming procedure in manual labeling, as shown in Figure 3 (c). Besides, this stage also needs to provide a category confidence for each detection box.

As shown in Figure 2, we use cascaded (3 for HumanLiker) tiny convolution blocks [1] to gradually improve the quality of the output bounding box. In each block, we apply both regression and classification like previous two-stage detectors [9, 27, 1]. For regression part, we predict four compensate values to further refine both top-left and bottom-right corner locations as:

$$\begin{cases} \tilde{\delta x} = (x^* - x)/\Delta x; \ \ \tilde{\delta y} = (y^* - y)/\Delta y \\ \tilde{\Delta x} = \log(\Delta x^*/\Delta x); \ \ \tilde{\Delta y} = \log(\Delta y^*/\Delta y) \end{cases} \tag{10}$$

where $(\tilde{\delta x}, \tilde{\delta y})$ and $(\tilde{\Delta x}, \tilde{\Delta y})$ are used to refine coordinates of the top-left and bottom-right corners. $\delta$ and $\Delta$ represent adjust relative less and more, respectively. With the above regressed compensate values, we can decode each adjusted corner coordinate as:

$$\begin{cases} x_{tl} = x + \tilde{\delta x} \cdot \Delta x; \ \ y_{tl} = y + \tilde{\delta y} \cdot \Delta y \\ x_{br} = x_{tl} + \exp(\tilde{\Delta x}) \cdot \Delta x; \ \ y_{br} = y_{tl} + \exp(\tilde{\Delta y}) \cdot \Delta y \end{cases} \tag{11}$$

In the training stage, the positive sample settings and loss function $\mathcal{L}_{ii}$ (including both classification and regression) of the second stage is following vanilla Cascade R-CNN [1]. During the inference, we define the detect box score as the geometric mean of the locate score and class score:

$$score = \sqrt{s(x, y) \cdot s(class)_m} \tag{12}$$

where the $s(x, y)$ is a locate score output in the first stage (Eq. 9) and $s(class)_m$ represents the category score which is the arithmetic mean of $s(class)_{1,2,...,n}$ output in each block. Through these definitions, each box score output by HumanLiker embeds both the location and class information.

Finally, the overall training objective of HumanLiker is defined as follows:

$$\mathcal{L}oss = (\lambda \mathcal{L}_h + \mu \mathcal{L}_o + \nu \mathcal{L}_r)_i + \mathcal{L}_{ii} \tag{13}$$

where $i$ and $ii$ represent the first and second stages, respectively. $\lambda$, $\mu$, and $\nu$ are three balance weights that set to 1, 2, and 0.001 following previous works [14, 35, 7].

# 4 Experiments

## 4.1 Datasets and Evaluation Metrics

We evaluate the proposed HumanLiker on the large-scale object detection benchmark — MS-COCO [21]. It contains 80 categories and more than $1.5$ million object instances. We train on the train2017 set which contains $118k$ images and $860k$ instances and compare the performance of HumanLiker with state-of-the-art methods on the test-dev set ($20k$ images) via the online evaluation server. All ablation studies are performed on the val2017 set that contains $5k$ images and $36k$ objects.

We utilize AP/FPS as the accuracy/efficiency evaluation metrics on the MS-COCO dataset. AP represents the average precision rate, which is a function of both recall and precision and computed over ten different IoU thresholds (*i.e.,* $50\%:5\%:95\%$) with all categories. FPS means frames-per-second, which is a commonly used metric to measure the inference speed of a model.

## 4.2 Implementation Details

In the training stage, we follow the commonly used settings in object detection to train the Human-Liker. To be specific, most of our models are trained on 4 RTX 3090 GPUs with a batch-size of 16 under the SGD optimizer for 24 epochs using the ImageNet [13] backbone initialization, if not otherwise specified. We apply a learning rate of $0.02$ for the first 16 epochs and then decay it by $\times 10$ at the $16$th and $22$th epoch. Specially, the model with SwinTransformer (large) is trained on 8 GPUs with a batch-size of 8 for 36 epochs upon the AdamW optimizer. We adopt the multi-scale training strategy that the shorter side of each input image is randomly selected from a range of $[480, 960]$.

During the inference stage, we use a Titan Xp or 3090 GPU to test the inference speed. For the single-scale testing, we resize each image with a shorter side of $640$, while for the multi-scale testing, we resize each image to a shorter side range of $[400: 200: 1400]$. In the post-processing, we run the NMS with a threshold of $0.6$ to filter out the redundant results.

## 4.3 Comparisons with State-of-the-art Detectors

We evaluate the performance of HumanLiker by comparing it with state-of-the-art detectors on MS-COCO test-dev. As shown in Table 1, HumanLiker achieves an AP of 50.2% under ResNeXt-101 ($32\times$8d) backbone in single-scale testing, which is a state-of-the-art accuracy for this backbone under the same settings. As stated in Section 1 and 3, HumanLiker avoids cumbersome center search and corner grouping like our humans in labeling. As a result, HumanLiker lives up to expectations to bring significant improvements in the corresponding indicators. Specifically, compared with the corner-guided CentripetalNet [6] which enjoys current optimal corner grouping algorithm, HumanLiker lifts 4.9% on $AP_{50}$ (from 63.0% to 67.9%) due to corrects a number of FPs generated by mismatched corners via a grouping-free manner. Besides, compared with the center-guided Deformable DETR [37], HumanLiker still achieves higher AP (50.2% *vs.* 50.1%) in the case of its $AP_{50}$ is 1.8% lower than that of Deformable DETR. Because the AP in COCO is defined as the mean of accuracy in low/high IoUs. We have reason to believe that the accuracy under higher IoUs (*e.g.,* $AP_{75:95}$) of HumanLiker surpasses the Deformable DETR in a large margin, which means our model is easier to yield higher-quality detection boxes by finding the top-left corner location directly. Moreover, compared with mature two-stage detector Cascade R-CNN [1], HumanLiker lifts the $AP_{75}$ by 1.9% (52.9% *vs.* 54.8%), which further bolsters our claim that directly estimating corners is better for pinpointing the boundary of an object than centers. Of course, the comparison with Cascade R-CNN also proves that the excellent performance of our model does not come from the two-stage structure, but from the human-like design idea. Furthermore, it is worth noting that the AP in single-scale testing of HumanLiker can rise to 53.8% under SwinTransformer (large) backbone only with 36 epochs. And you only need 8 3090 GPUs to train the large SwinTransformer-based HumanLiker within 4 days, which indicates the HumanLiker is easy to converge and friendly to follow. All experimental results in Table 1 demonstrate that the proposed HumanLiker is powerful and promising to become an excellent baseline in the further.

Table 1: State-of-the-art comparisons in term of accuracy (%) with different object detectors on COCO test-dev. In this table, the ResNeXt-101 backbone of all models is the $32\times8$d version for a fair comparison. $*$ represents multi-scale testing. The abbreviations of "DCN"and "L" represent the Deformable Convolutional Networks [5] and "Large", respectively.

| Method (test-dev set) | Backbone | Epoch | AP | $AP_{50}$ | $AP_{75}$ | $AP_S$ | $AP_M$ | $AP_L$ |
|---|---|---|---|---|---|---|---|---|
| **corner-guided:** | | | | | | | | |
| CornerNet [14] | Hourglass-104 | 200 | 40.5 | 56.5 | 43.1 | 19.4 | 42.7 | 53.9 |
| CenterNet [7] | Hourglass-104 | 190 | 44.9 | 62.4 | 48.1 | 25.6 | 47.4 | 57.4 |
| CentripetalNet [6] | Hourglass-104 | 210 | 45.8 | 63.0 | 49.3 | 25.0 | 48.2 | 58.7 |
| **center-guided:** | | | | | | | | |
| Faster R-CNN [27] | ResNet-101 | 24 | 36.2 | 59.1 | 39.0 | 18.2 | 39.0 | 48.2 |
| RetinaNet [20] | ResNet-101 | 36 | 40.8 | 61.1 | 44.1 | 24.1 | 44.2 | 51.2 |
| CenterNet [36] | Hourglass-104 | 200 | 42.1 | 61.1 | 45.9 | 24.1 | 45.5 | 52.8 |
| DETR [3] | ResNet-101 | 500 | 44.9 | 64.7 | 47.7 | 23.7 | 49.5 | 62.3 |
| Reppoint [33] | ResNet-101-DCN | 24 | 45.0 | 66.1 | 49.0 | 26.6 | 48.6 | 57.5 |
| FCOS [30] | ResNeXt-101-DCN | 24 | 46.6 | 65.9 | 50.8 | 28.6 | 49.1 | 58.6 |
| ATSS [34] | ResNeXt-101-DCN | 24 | 47.7 | 66.6 | 52.1 | 29.3 | 50.8 | 59.7 |
| GFLv1 [17] | ResNeXt-101-DCN | 24 | 48.2 | 67.4 | 52.6 | 29.2 | 51.7 | 60.2 |
| Cascade R-CNN [1] | ResNeXt-101-DCN | 24 | 48.8 | 67.7 | 52.9 | 29.7 | 51.8 | 61.8 |
| Sparse R-CNN [29] | ResNeXt-101-DCN | 36 | 48.9 | 68.3 | 53.4 | 29.9 | 50.9 | 62.4 |
| GFLv2 [16] | ResNeXt-101-DCN | 24 | 49.0 | 67.6 | 53.5 | 29.7 | 52.4 | 61.4 |
| RepPointv2 [4] | ResNeXt-101-DCN | 24 | 49.4 | 68.9 | 53.4 | 30.3 | 52.1 | 62.3 |
| Deformable DETR [37] | ResNeXt-101-DCN | 50 | 50.1 | 69.7 | 54.6 | 30.6 | 52.8 | 65.6 |
| **human-like:** | | | | | | | | |
| HumanLiker | ResNeXt-101-DCN | 24 | 50.2 | 67.9 | 54.8 | 30.7 | 53.9 | 63.8 |
| HumanLiker$*$ | ResNeXt-101-DCN | 24 | 51.6 | 68.4 | 57.1 | 34.2 | 55.0 | 64.3 |
| HumanLiker | SwinTransformer-L | 36 | 53.8 | 72.2 | 58.8 | 35.3 | 57.0 | 68.3 |
| HumanLiker$*$ | SwinTransformer-L | 36 | 55.6 | 72.5 | 61.5 | 39.1 | 59.1 | 69.4 |

## 4.4 Comparisons on Inference Speed

We evaluate the inference speed of HumanLiker on MS-COCO val2017 set. As shown in Table 2, HumanLiker under ResNet-50 can achieve an AP of 43.9% at a 15 FPS on a Titan Xp GPU with only 24 training epochs. HumanLiker yields on-par inference speed with Faster R-CNN or Sparse R-CNN (15 *vs.* 16 *vs.* 14 FPS) while achieves higher accuracy (43.9% *vs.* 40.2% *vs.* 42.8% AP) with less training epochs (24 *vs.* 36 *vs.* 36). Besides, equipped with the SwinTransformer (tiny) and 3090 GPU, HumanLiker can achieve a better trade-off in terms of accuracy (47.1% AP) and efficiency (18 FPS). It is worth repeating that the quality of detection boxes of HumanLiker is quite high: Only with ResNet-50, HumanLiker can report an $AP_{75}$ of 48.3% which is even higher than that of other models (*e.g.,* DETR, and Sparse R-CNN) along with the ResNet-101. On balance, there are almost no short boards for HumanLiker to become a splendid baseline, no matter in terms of accuracy, inference speed, and training convergence speed.

Table 2: Comparisons with a few classic detectors in term of inference speed on MS-COCO val2017. The abbreviations of "R50", "R101", and "Swin(t)" represent different backbones, *i.e.,* ResNet-50, ResNet-101, and SwinTansformer (tiny), respectively.

| Method | GPU | FPS | Epoch | AP | $AP_{50}$ | $AP_{75}$ | $AP_S$ | $AP_M$ | $AP_L$ |
|---|---|---|---|---|---|---|---|---|---|
| RetinaNet-R50 | Titan Xp | 12 | 36 | 38.7 | 58.0 | 41.5 | 23.3 | 42.3 | 50.3 |
| Faster R-CNN-R50 | Titan Xp | 16 | 36 | 40.2 | 61.0 | 43.8 | 24.2 | 43.5 | 52.0 |
| DETR-R50 | Titan Xp | 7 | 500 | 43.3 | 63.1 | 45.9 | 22.5 | 47.3 | 61.1 |
| DETR-R101 | Titan Xp | 5 | 500 | 44.9 | 64.7 | 47.7 | 23.7 | 49.5 | 62.3 |
| Sparse R-CNN-R50 | Titan Xp | 14 | 36 | 42.8 | 61.2 | 45.7 | 26.7 | 44.6 | 57.6 |
| Sparse R-CNN-R101 | Titan Xp | 10 | 36 | 44.1 | 62.1 | 47.2 | 26.1 | 46.3 | 59.7 |
| HumanLiker-R50 | Xp/3090 | 15/**23** | 24 | 43.9 | 60.6 | **48.3** | 26.4 | 47.7 | 57.2 |
| HumanLiker-Swin(t) | Xp/3090 | 12/18 | **12** | **47.1** | 65.0 | 51.6 | 30.5 | 51.0 | 61.8 |

## 4.5 Ablation Study

**Top-left corner *vs.* center.** In the Section 1, we claim that a top-left corner is easier to be located than a center for our humans and this is a consensus in manual labeling. To verify if the claim works for a network, we design two models that only estimate top-left corners or centers, respectively. We conduct experiments to observe their loss values in training to judge which keypoint location is easier to learn. The keypoint estimate strategy for each model is the same as mentioned in Section 3.3.1. As shown in Figure 4, under exactly the same setting, loss values of the top-left corner model is lower than that of the center model in the prophase of training, demonstrating that the top-left corner location is also easier to learn than a center for a detection model.

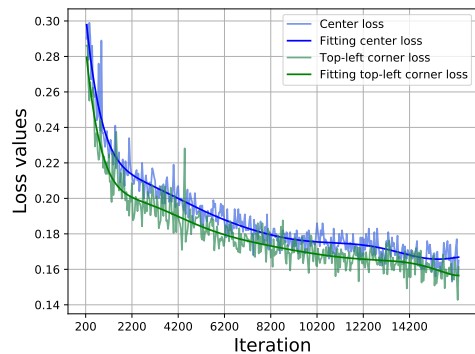

Figure 4: Training loss values comparison between top-left corner and center estimation models.

Table 3: Effectiveness of each design in HumanLiker on COCO val2017 set. All experiments are conducted under SwinTransformer (tiny) backbone along with 12 epochs training on the 3090 GPU. "CP" and "CCA" represent corner pooing and criss-cross attention used in state-of-the-arts.

| Offset | Stage two | Cascade | CP | CCA | AP | $AP_{50}$ | $AP_{75}$ | FPS |
|--------|-----------|---------|----|----|----|-----------|-----------|-----|
| ✓ | ✓ | ✓ | × | × | 47.1 | 65.0 | 51.6 | 18 |
| × | ✓ | ✓ | × | × | 46.7 | 64.5 | 51.1 | 18 |
| ✓ | × | ✓ | × | × | 45.4 | 63.5 | 50.6 | 18 |
| ✓ | ✓ | × | × | × | 46.2 | 64.1 | 50.9 | 20 |
| ✓ | ✓ | ✓ | ✓ | × | 47.1 | 64.9 | 51.7 | 15 |
| ✓ | ✓ | ✓ | × | ✓ | 47.0 | 64.7 | 51.5 | 16 |

**Effectiveness of the offset.** We conduct this experiment to show if the top-left offset design in HumanLiker is efficient. As shown in Table 3, the offset adjustment in the first stage gains 0.4% AP improvement (from 46.7% to 47.1%), proving such design is resultful. The reason for the small increase is that HumanLiker will also refine the top-left locations in the second stage.

**Effectiveness of the second stage.** We remove the fine trimming (regression) branch of the stage two to verify the effectiveness of refining corner coordinates and the AP drops 1.7% (form 47.1% to 45.4%). The result demonstrates that the human-like two-stage design is momentous.

**Effectiveness of cascade blocks.** We keep one block in the stage two of HumanLiker to test the effectiveness of multiple cascade blocks. We can see the inference speed of HumanLiker gains 2 FPS faster (form 18 to 20) yet the AP obtains a drop of 0.9% (form 47.1% to 46.2%). The result indicates that the cost performance of cascade blocks is relatively high.

**The corner feature enhancement.** We do not use any corner feature enhancement modules, *e.g.*, corner pooing [14] or criss-cross attention [12], in HumanLiker. In Table 3, we find that they have no effect on our HumanLiker. The reason may be that HumanLiker adopts multi-level features, which is enough for predicting top-left corners of multiple sizes objects, especially for large ones.

## 4.6 Improvement Room

We believe there are three main improvement directions of HumanLiker, namely, 1) False estimated top-left corners generated due to boundary confusion. In Figure 5 (a), the corner in the "red dotted circle" is composed of boundaries belong to two different persons (number "9" and "26"). 2) Imprecise estimated top-left corner due to boundary occlusion. In Figure 5 (b), the two "cows" coincide visually, causing the corner of the shaded one to be imprecise. 3) In Figure 5 (c), due to the difficulty of long-range regression, the predict bottom-right corner is not as precise as the top-left one ("zoom in" for details), harming the further improvement of the box quality.

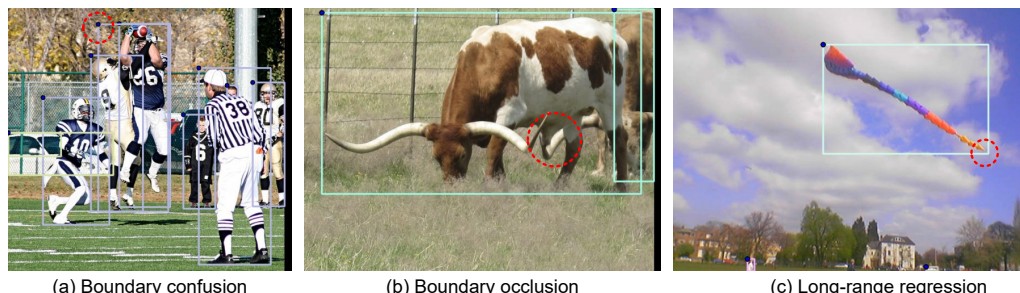

| (a) Boundary confusion | (b) Boundary occlusion | (c) Long-range regression |

Figure 5: Three main improvement directions of the proposed HumanLiker. (a) Different instances with same class may form a false top-left corner. (b) Occlusion conditions may cause inaccurate estimates of the top-left corner. (c) The accuracy of the bottom-right corner has room for purifying.

## 4.7   Visualization Results

We provide high-quality visualization detection results, generated on MS-COCO val2017, of the proposed HumanLiker to show its performance visually. As shown in Figure 6, we can see the boundary of each detection box can fit well with the object, yielding a good $AP_{75}$.

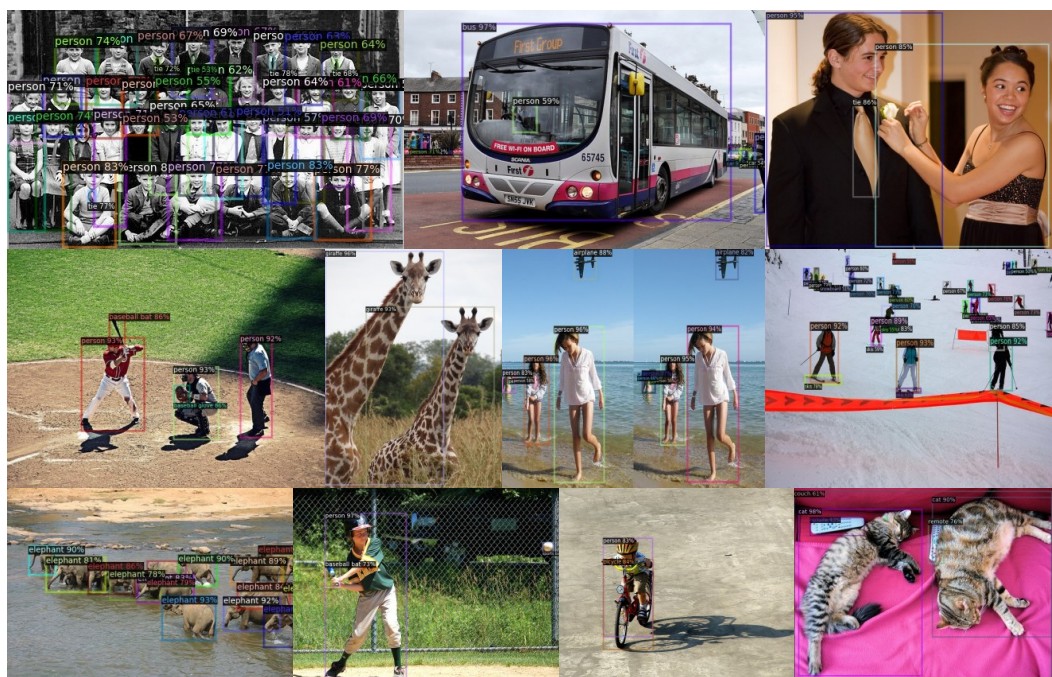

Figure 6: Some qualitative detection results on the MS-COCO validation dataset.

## 5   Conclusion

In this paper, we deeply analyze the gap between detection algorithms and our humans in generating object boxes. Based on our analysis, we propose a novel human-like detection paradigm along with a detector, called HumanLiker, to further explore the potential of deep-network-based detectors. Experimental results show that the HumanLiker enjoys immense advantages in terms of accuracy, inference speed, and training convergence speed. We believe that the HumanLiker is able to become a promising baseline in the further. More importantly, our research reveals that much room is left in constructing object detector from human-like cues. We hope the simple and efficient design of HumanLiker will attract more attention to devise detection models from a bionic perspective.

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
