# OpenReview forum: "HumanLiker: A Human-like Object Detector to Model the Manual Labeling Process"
_NeurIPS.cc/2022/Conference — NeurIPS 2022 Accept_

### Official Review · Reviewer_g27y · 2022-07-08

**Rating:** 4
**Confidence:** 5
**Soundness:** 2 fair
**Presentation:** 3 good
**Contribution:** 2 fair

**Summary:**

In this paper, the authors study the problem of object detection. To be specific, they introduce a new approach for bottom-up detection of objects in images. They take inspiration from how humans label objects (click top-left corner, drag to the bottom-right corner) and propose a two-stage approach mimicking this. They report positive results on the COCO dataset.

**Questions:**

(1) More justifications are require to call this approach human like.

(2) More controlled experiments are required to justify the source of improvement over CornerNet.

**Limitations:**

The method is neat, powerful, intelligible, bionic. It doesn't have any limitations.

**Strengths And Weaknesses:**

Strengths:
+ Taking inspirations from humans in labeling objects is very interesting.

+ A novel mechanism is introduced that mimicks human labeling.


Weaknesses:

- The paper talks about how humans label objects and makes certain assumptions about the process but these are based on beliefs and observations of the authors. One can easily argue that humans might look at/around object centers to determine the top-left corner and then the bottom-right corner. It is not clear whether top-left => bottom-right ordering is affected by right-handedness or cultural differences.

- The paper's improvement over the existing corner-based methods is not clearly demonstrated in the experiments. HumanLiker has a second stage, DCN and therefore extra processing compared to e.g. CornerNet and this can have a huge impact. A more controlled evaluation is needed to justify the contribution of the paper.

- Line 71: "neat and bionic, which makes the model intelligible and friendly to follow." => How do you define and measure these? It is not clear how the proposed approach is related to bionics.

- "2) Instead of single-level feature, we use multi-level features based on FPN [19] to better fit object corners with different size and context;" => Is this really a contribution?

- The proposed mechanism has too many hyper-parameters. Considering the complicated two-stage approach, I wouldn't call this neat or intelligeble. And I would avoid using unnecessarily positive and/but unjustified adjectives such as powerful, bionic, neat, ..

- Fig 4: The difference is so small between the two lines that I am not sure it is safe to make a conclusion from here. With N different random runs, you might obtain different outcomes.


Minor comments:
- Line 39: "for object with peculiar geometry" => "for objects with peculiar geometry".
- Line 158: "which close to the positive" => "which are close to the positive".
- Eq 1: "Objet size" => "Object size". Gk should be defined as a function which takes some input.
- Eq 2: space after "if". "xy" => "x,y".
- Line 164: You should cite on this line CornerNet for Eq 2. The reference on line 167 is insufficient & indirect for the source of Eq 2.
- Eq 2: Please explicitly state that this is class agnostic.
- Line 174: "Due to there" => "Due to the fact that there".
- Line 180: "offset." => "offset:".
- Please read the following guide about writing equations: http://www.ai.mit.edu/courses/6.899/papers/mermin.pdf
- Line 203: "detect box." => "detection box."
- Eq 13: Please use different symbols for these hyper-parameters as you used them before.
- L_ii should be explicitly written in the paper for completeness.

---

> ### Author Response · Authors · 2022-08-02
> **Reply item by item for Reviewer g27y**
>
> Thank you very much for your careful review and valuable comments. We address your concerns as follows:
>
> 1. This question is interesting. However, we believe your hypothesis that humans may deduce the top-left corner and the bottom-right corner inversely through the center of the object is not tenable. It is because that pinpointing a center needs all four boundaries as reference and a corner only needs two boundaries.  This is a strong prior knowledge that locating a corner of an irregular object is easier than a center.  Besides, the top-left => bottom-right ordering or a bottom-right => top-left  doesn't affect our ‘human-like’ claim because we describe: manual labeling avoids center searching or corner grouping due to high degrees of freedom in searching centers or low efficiency of grouping corners. A top-left => bottom-right ordering in our HumanLiker is just to represent the habits of most people.
>
>
> 2. The newest corner-based detector CPN also has a second stage but its accuracy, inference speed, and convergence speed are lower than our HumanLiker. For CornerNet and its varients (CenterNet and CentripetalNet), the Hourglass-104 backbone is very friendly for corner estimation. Replacing the Hourglass-104 with ResNeXt-101+DCN for them suffers 2~3% AP drop. Besides, HumanLiker is grouping-free, we try to adapt the corner estimation/grouping method of CPN (a binary sub-network) and CornerNet (associative embeddings) to our sturcture (FPN/category unknowable heatmap/two-stage refinement/classification) to further control variables. Due to they need corner grouping, for each FPN layer, we optimized the number of corners (64,32,16,8,4) to be extracted through ablation experiments for each layer. Experiments show that the CPN and CornerNet drop about 4% and 7% (from 47.1% to 43.2% and 40.4%) AP than our HumanLiker on Swin-T backbone for 1x epoch, proving the proposed HumanLiker is better than other corner-based detectors.
>
> 3. We are sorry for the misleading use of words. The bionic we mean is human-like.  The neat, intelligible, and friendly we think they are for an object detection beginner. A beginner can easily understand at a macro level what each module does and the process that corresponds to manual labeling. Of course, we will carefully consider and be willing to replace these inappropriate words in the next version of manuscript.
>
> 4. Traditional corner detection model needs to predict corner via outputing heatmap.  They always use a large (x4 output stride) and multi-channels (80 for 80 classes) heatmaps to predict corners. Adapting FPN will encounter several problems: (a) More heatmaps (channels) make corner extraction suffer lower efficiency. (b) More predicted corners make corner grouping more difficult. Our HumanLiker only needs one heatmap channel and runs in a grouping-free way so that it is suitable for FPN. Accordingly, we believe the sucess of using FPN for corner-based detector is important and valuable.
>
> 5. We think the neat and intelligeble describe are  facing object detection beginners at a macro level understand. The beginner can compare his labeling process to understand the design of each module of the HumanLiker. We really appreciate your guidance in writing and we'll use more appropriate words and detailed descriptions in the next manuscript.
>
> 6. We change different backbones (ResNet-50,101,DLA-34, Hourglass-52) to draw the training loss of center and top-left corner estimation and all the trends are similar to Figure 4: the top-left corner loss is  lower than center. Besides, we sample the AP at the 0.5x，1x，1.5x，and 2x epoch. HumanLiker achieves 40.1%, 41.8%, 43.0%, and 43.9% AP and the new center-guided model (change the top-left corner to center) yields AP of 39.5%, 41.3%, 42.7%, and 43.7% on ResNet-50 backbone, which further proves our claim that a corner is easier to predict than a center especially at the beginning of training is reasonable. This is also easy to understand because a center needs all four boundaries to locate yet a corner only needs two.
>
> 7. Typos and equations: Thanks for these useful comments! We will carefully revise our manuscript.

---

> > ### Comment · Reviewer_g27y · 2022-08-04
> > **Thank you**
> >
> > Dear authors,
> >
> > Thank you for the detailed responses which address my concerns.
> >
> > Best

---

### Official Review · Reviewer_MgUj · 2022-07-08

**Rating:** 5
**Confidence:** 5
**Soundness:** 3 good
**Presentation:** 3 good
**Contribution:** 3 good

**Summary:**

This paper proposes a new object detector based on CornerNet. The proposed detector adopts a two-stage design. In the first stage, instead of detecting both top-left and bottom-right corners, and grouping them by embeddings, the authors propose to detect only the top-left corners and predict the x and y distances between the top-left corners and bottom-right corners to generate an initial set of regions. In the second stage, instead of refining the width and height of the bounding boxes, they propose to refine the corner locations and the distances. The authors argue the this is closer to how human would draw a bounding box. Experiments on COCO show that the proposed detector achieves state-of-the-art results with same backbone.

**Questions:**

My main concern is that the current experiments do not sufficiently show that the proposed approach is better than the existing approaches. How does regressing distance compare to associative embeddings or corner proposal network in terms of validation performance? How does detecting corners compare to detecting centers in terms of validation performance? How does refining corner location and distances compare to the conventional refining strategy in terms of validation performance? Overall I like the idea of this paper so if the authors provide satisfactory answers to the above questions, I am happy to reconsider my rating.

**Strengths And Weaknesses:**

Strengths:

The proposed approach is interesting and novel, and demonstrates good results on the challenging COCO benchmark. I like the idea of refining corner locations instead of the whole box in the second stage.

Weaknesses:

The main weakness of this paper is that the current experiments do not show sufficiently show that the proposed approach is better than the conventional approach.

The network generates the proposals by predicting the top-left corners, and x and y distances between top-left corners and bottom-right corners. This is different from the corner proposal network [8] or the associative embeddings used in CornerNet. But there is no ablation study that compares these three approaches.

The authors claim that it is easier to learn to detect corners than the centers by showing that the top-left corner model achieve a lower training loss than the center-based model. There are two problems with this claim. First only comparing the difficulties of detecting corners and centers is not sufficient because there are other predictions that are used to generate the final bounding boxes, which may be difficult to predict. Second, training loss is not a good indicator of the final performance so it would be better to compare the models by their validation performance.

In the second stage, the authors propose to refine the corner locations and distances where a conventional object detector would directly refine the center locations. But there are no experiments that support refining corner locations and distances is better and more accurate.

Although the experiments show that the proposed approach outperforms other detectors with the same backbone network, it is not clear where the improvement comes from because the proposed approach uses a cascade network which is known to improve performance while others don’t.

---

> ### Author Response · Authors · 2022-08-02
> **Reply item-by-item for Reviewer MgUj**
>
> Thank you very much for your valuable and professional comments！We are sorry that we did not report more experimental results and details in the previous version of manuscript. This is due to the limitation of the number of pages and we are very happy to add a page to supplement these contents if this paper can be accepted fortunately.  We address your concerns below.
>
> 1. Comparison with CPN and CornerNet: CPN and CornerNet are two grouping-based model yet HumanLiker is grouping-free. To group corner pairs more effective, CPN and CornerNet need to assign corners of different categories of objects to different heatmap channels (80 for top-left and 80 for bottom-right), leading to the corner extraction process suffer low efficiency. HumanLiker, with grouping-free manner, only needs one category unknowable heatmap channel for top-left corners. The FPS is 21.7/9.9/5.8 for HumanLiker/CPN/CornerNet under 43.9%/43.8%/41.0% AP on V100 GPU. What's more, the training epoch of HumanLiker is 5x to 10x less than CPN and CornerNet, which shows our HumanLiker is easier to converge.  More improtantly, the grouping-free design enjoys better accuracy: We try to adapt  the corner estimation/grouping method of CPN (a binary sub-network) and CornerNet (associative embeddings) to our sturcture (FPN/category unknowable heatmap/two-stage refinement/classification). Due to they need corner grouping, for each FPN layer, we optimized the number of corners (64,32,16,8,4) to be extracted through ablation experiments  for each layer. Experiments show the CPN and CornerNet drop about 4% and 7% (from 47.1% to 43.2% and 40.4%) AP than our HumanLiker on Swin-T backbone for 1× epoch, proving the HumanLiker is stronger than other corner-based detectors firmly.
>
> 2. Center vs. Corner: Thanks for your professional comment. We use the comparision of training loss to show the strong prior knowledge of  manual labeling that pinpointing a corner is easier than a center is also true for a model. Based on your said, we use the center and width/height (x_c,y_c,w,h) to replace our design in the stage one of HumanLiker to prove our claim. We sample the AP at the 0.5x，1x，1.5x，and 2x epoch. HumanLiker achieves 40.1%, 41.8%, 43.0%, and 43.9% AP and the new center-guided model yields AP of 39.5%, 41.3%, 42.7%, and 43.7% on ResNet-50 backbone. The experimental results also prove our claim that a corner is easier to predict than a center especially at the beginning of training. This is also easy to understand because a center needs all four boundaries to locate yet a corner only needs two.
>
> 3. The refinement in the stage two:  We use a corner decoupling strategy (refine each corner separately)  to refine each corner of a proposal box. Traditional models utilize a corner coupling (center is calculated via two corners) way. Suppose one boundary of an object is difficult to predict, the center which needs four boundries as reference is also hard to refine. However, corner decoupling (refine each corner separately) can make sure that a corner is not affected by this boundry and yield higher quality results (e.g., better AP_80 to AP_90). The AP_70,80,90 on COCO val of HumanLiker are 55.9%, 45.5%, 26.1%. On contrast, when we use the traditional center refinement on HumanLiker, the AP_70,80,90 are 55.8%, 45.1%, 25.6%. The results shows that our human-like refinement strategy is easier to provide high-quality detection box.
>
>
> Finally, we respectfully emphasize that the main motivation of HumanLiker is to provide a new detector design idea for detection community.  As we know, center-guided model is mature and popular but they also suffer a bottleneck. HumanLiker is devised as a new baseline enjoying  promising  improvement room to inspire researers that use a corner as a positive sample like our humans can also work well.

---

### Official Review · Reviewer_rgaq · 2022-07-10

**Rating:** 5
**Confidence:** 5
**Soundness:** 3 good
**Presentation:** 3 good
**Contribution:** 2 fair

**Summary:**

This paper is inspired by the human way of labeling object boxes in an image. It proposes a two-stage detector:

- First, the detector will localize the top-left corner of an object and regress the height and width of the object.
- Then, the detector will refine the box using the cascaded method.

The method is verified on MS-COCO test-dev with multiple backbones, e.g., ResNetXt-101 + deformable convolutional module and SwinTransformer-L.


**Questions:**

- YOLO-based methods didn’t show up in the big Table.1. It could be better to add more related top-ranking detectors in it.

**Limitations:**

None.

**Strengths And Weaknesses:**

Strengths:
- Writing is pretty well-formulated and well-organized. It’s easy to follow.
- The proposed framework belongs to the two-stage one but is different from the conventional two-stage detector. It doesn’t have the region proposal idea in it. Instead, it is developed based on the key-point and grid box regression. The second stage aims to refine the box regression.
- The experimental results demonstrate that the method is promising with large improvements.

Weakness:
- Current training strategy for object detection becomes large-scale jittering and fixed input size (1024 x 1024). The training way in the paper is a little outdated. I understand if changing the training method will make comparisons more difficult. So I suggest the paper could use the new training way without comparisons to produce stronger results. This would ease future comparisons in the detection community.

---

> ### Author Response · Authors · 2022-08-02
> **Reply for Reviewer rgaq**
>
> Thank you very much for your valuable and professional comments.
> As you said, for a more fair comparison, we didn't report the AP under a large and fixed input size in the previous version of manuscript. We conducted experiments on Res2Net-101 with fixed size (896x896 and 1024x1024) for 4x training epochs. The APs (on COCO val2017) are 51.4% and 52.6%. The experiment results show that a fixed and  large input size does improve the accuracy of HumanLiker.
> Besides, we are more than happy to add the fixed input size and YOLO-based SOTAs for comparision in the next version of manuscript.

---

### Meta-Review · Area_Chair_1ytP · 2022-08-26

**Recommendation:** Accept
**Confidence:** Certain

**Metareview:**

This paper received borderline reviews, with one review leaning negative. However, the reviewer acknowledged that their concerns have been addressed but did not update the rating.

The paper provides an interesting new take on object detection with strong empirical results. The concerns raised by reviewers were mainly about more experimental results and clarifications, which the authors have adequately addressed in their rebuttal.

For the camera ready version, the authors must change the title of the paper to something more informative. "A human-like object detector" is too vague and non-specific, and misleading given the paper's actual contribution. "human-like" can mean many things (e.g. learning  from very few examples), but the paper is only "human like" in one aspect. In addition, "humanliker" should be replaced by something else ("humanliker" can mean "something that likes human").

**Award:**

No

---

### Decision · Program_Chairs · 2022-09-14

Accept